# Change Detection for bi-temporal images classification based on Siamese Variational AutoEncoder and Transfer Learning

## Abstract

Siamese structures empower Deep Learning (DL) models to increase their efficiency by learning how to extract the relevant temporal features from the input data. In this paper, a Siamese Variational Auto-Encoder (VAE) model based on transfer learning (TL) is applied for change detection (CD) using bi-temporal images. The introduced method is trained in a supervised strategy for classification tasks. Firstly, the suggested generative method utilizes two VAEs to extract features from bi-temporal images. Subsequently, concatenates them into a feature vector. To get a classification map of the source scene, the classifier receives this vector and the ground truth data as input. The source model is fine-tuned to be applied to the target scene with less ground truth data using a TL strategy. Experiments were carried out in two study areas in the arid regions of southern Tunisia. The obtained results reveal that the proposed method outperformed the Siamese Convolution Neural Network (SCNN) by achieving an accuracy of more than 98%, in the source scene, and increased the accuracy in the target scene by 1.25% by applying the TL strategy.

## 1 Introduction

The feature extraction step in the classification process allows improving DL model performance in several fields (Hakak et al., 2021; Islam & Nahiduzzaman, 2022; Xiong & Zuo, 2022). In fact, Convolutional neural network (CNN) has been efficiently employed to solve computer vision problems in a variety of fields including industry, environment, and healthcare (Alzubaidi et al., 2021; Huang et al., 2022). Nevertheless, the performance of the algorithms depends of the used datasets. Furthermore, CNN has shown a low performance in the classification task thanks to the high similarity and non-dispersity of the input data. Recently, with these challenging, the VAE has demonstrated its good performance in the classification tasks as it is based on distribution-free assumptions and nonlinear approximation (Zerrouki et al., 2020; Ran et al., 2022). However, the periodicity of the input data reduces its efficiency and, therefore, makes it unable to ensure the temporal consistency of the extracted features (Zhao & Peng, 2022). Moreover, traditional DL models (e.g. CNN, VAE, etc.) cannot capture the temporal information. Thus, they have limited capability to extract the temporal features. To overcome this shortcoming, the Siamese structure, which is one of the best approaches for CD in bi-temporal images, can be a good solution. Siamese networks were first utilized for signature verification. Subsequently, they were applied in feature matching, particularly between pairs of images (Ghosh et al., 2021; Zhang et al., 2022). Recent studies focusing on classification tasks have employed bi-temporal images for CD (Lee et al., 2021; Zheng et al., 2022). The CD process consists in identifying the differences between bi-temporal images of the same geographic location undergoing anthropic and climatic factors . Exploring the generalization of Siamese DL models is a key challenge. Discussing its TL capabilities is one of the most popular analyses (Krishnamurthy et al., 2021; Abou Baker et al., 2022). The TL aims at gaining knowledge by solving a problem and applying it to another related problem. The use of TL in practice is to apply knowledge from one context with several labeled data to another situation with limited labels. In application, TL consists in re-using the weight values of the trained model with source data, while applying a fine-tuning approach to provide a model adapted to the target data (Raffel et al., 2020; Shabbir et al., 2021; Toseef et al., 2022). By employing the pre-trained model source as the target scene adapter instead

of starting from the scratch, the fine-tuning technique reinforces learning and considerably reduces the model overfitting (Tan et al., 2018; Cao et al., 2022). The contributions of the present work are presented below:

• Proposing a new method for bi-temporal images classification based on Siamese VAE, in order to extract the relevant temporal features.

• Using a TL strategy to transfer the pre-trained Siamese VAE from the source to the target scene.

• Evaluating the introduced method w.r.t. SCNN, in two study areas, using bi-temporal multispectral images acquired with Landsat.

The rest of this manuscript is organized as follows. Some related works are described in section 2. The developed technique and the background of the Siamese VAE and the TL strategy used in this study are presented in section 3. Section 4 depicts the experimental settings, the implementation details and the applied evaluation metrics. The obtained results are provided and discussed in section 5, while Section 6 concludes the paper and gives some future perspectives.

## 2 RELATED WORK

Recently, numerous studies have focused on Siamese structure using bi-temporal images to enhance feature extraction-based classification models for CD. For example, Zhu et al. (2022) have designed a Siamese global learning (Siam-GL) framework for high spatial resolution (HSR) remote sensing images. The Siamese structure has been used to improve the feature extraction of bi-temporal HSR remote sensing images. Researchers have concluded that the Siam-GL framework outperformed the advanced semantic CD methods as it provided more pixel data and ensured high precision classification. Besides, Zhao & Peng (2022) have presented a semi-supervised technique relying on VAE with the Siamese structure to detect changes in Synthetic Aperture Radar (SAR) images by concatenating the extracted features of bi-temporal images in a single latent space vector to extract the pixel change characteristics. Moreover, Daudt et al. (2018) proposed a CD framework based on a Siamese CNN for CD. In the suggested method, Sentinel-2 multispectral pair images were encoded via a Siamese network to extract a new data representation. Then, the extracted bi-temporal representations were combined to produce an urban CD map. The designed network was trained in a fully supervised manner and showed excellent test performance.

Indeed, the performance of a DL model can be enhanced applying TL strategy, especially CD models based on Siamese structures. We list, in this paragraph, some research works relying on Siamese CD and TL. For instance, Yang et al. (2019) proposed a DL-based CD framework with TL strategy to apply a learned source CD model in a target domain. The introduced framework includes pre-training and fine-tuning steps, while the target domain is used to fit the source domain modification concept. A method of image difference in the target domain was utilized to pick pixels having a high probability of being correctly classified by an unsupervised technique, which improved the change detection network (CDN) of the target domain. The provided findings showed that the developed method outperforms the state-of-the-art CD techniques and offers an excellent ability to transfer the concept of change from the source domain to the target domain. Heidari & Fouladi-Ghaleh (2020) presented a face recognition platform based on TL in a Siamese network made up of two identical CNNs. On this platform, the Siamese network extracts the features from the input pair of images and determines whether they belong to the same person or not. The experimental results revealed that the accuracy of the proposed model (95.62%) is better than that of the state-of-the-art methods of face recognition. Bandara & Patel (2022) designed a transformer-based Siamese network architecture (ChangeFormer) for CD by considering a pair of co-registered remote sensing images. In the proposed Siamese network, convolutional networks (ConvNets) were used to transform the multiscale information into long-range data needed for CD. The experimental findings demonstrate that the developed end-to-end trainable architecture outperforms the existing ones as it enhances the CD performance. Andresini et al. (2022) suggested a Siamese network trained utilizing labeled imagery data of the same land scene acquired by Sentinel-2 at various times to detect changes in land cover in bi-temporal images. The trained Siamese network in a labeled scene was transferred to a new unlabeled scene applying a fine-tuned TL strategy. The lack of change labels in the new scene was addressed by estimating pseudo-change labels in an unsupervised manner. The experiment

conducted in Cupertino and Las Vegas showed that the proposed strategy performs better than the standard Siamese networks trained either in a supervised or unsupervised manner.

## 3 METHOD

We consider two bi-temporal image scenes with two different pixel grids having different spatial locations: $X_s$ as the source and $X_t$ as the target. We also take into account the matrices $C_s$ and $C_t$ containing information about the ground truth classes available for pixels in the scenes $X_s$ and $X_t$, respectively. Note that the size of $C_t$ is always smaller than that of $C_s$. The Siamese VAE method classifies the detected changes in two steps in a supervised manner. A Siamese network is trained, in the first step, using a supervised classification task depending on the type of CD method for bi-temporal image source scene $X_s$ and ground truth $C_s$. Then, the second step consists in deploying the trained Siamese VAE to the bi-temporal image target scene $X_t$ and using the labeled data of $C_t$ employing a fine-tuning approach. The schema of the proposed methodology is illustrated in Figure 3.

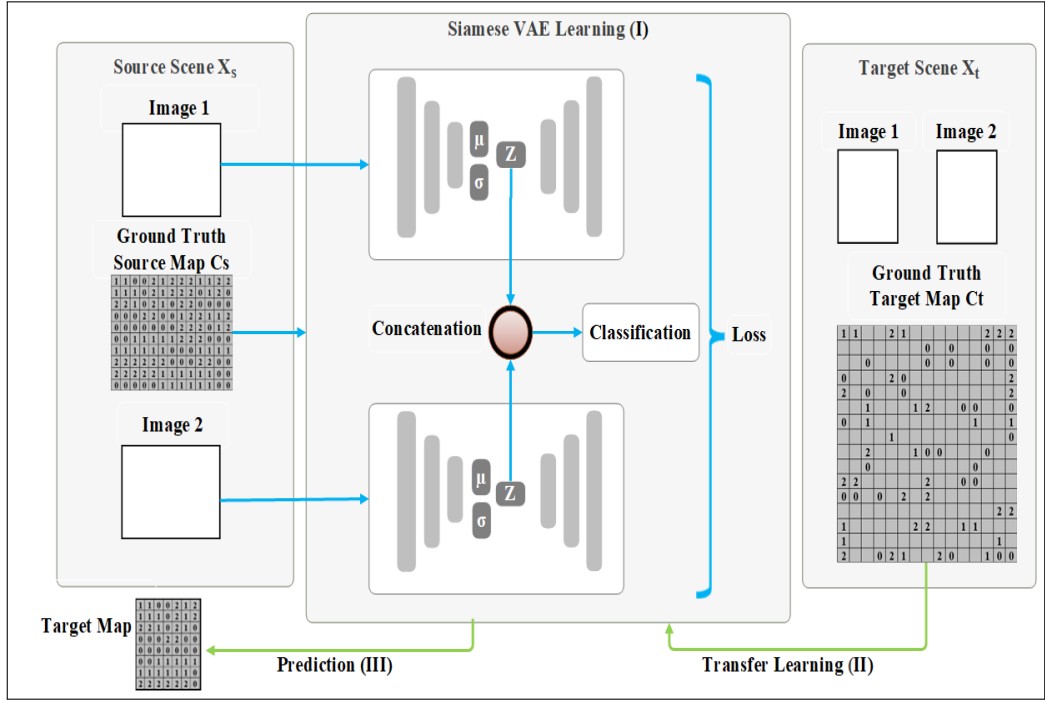

Figure 1: Flowchart of the proposed method: (I) Siamese VAE was used to extract the latent spaces from the source scene of bi-temporal images. These latent spaces are then concatenated to create a feature vector, which is used as input for the classifier trained on ground truth data. (II) A TL strategy was employed to improve the trained Siamese VAE on the bi-temporal source images. (III) The classification map of the target scene was predicted using the target Siamese VAE.

### 3.1 VARIATIONAL AUTO-ENCODER BACKGROUND

The VAE is made up of two components (Kingma & Welling, 2013), as indicated in Figure 3.1. The first one, called the encoder, is a clustering part that projects data input X into a latent space Z according to a Gaussian probability P(Z|X), with a regularization trick provided by a vector of means $\mu$ and a vector of standard deviations $\sigma$. However, the second component, named decoder part, generates the original data X as X' from the latent space Z according to a probability distribution Q(X|Z).

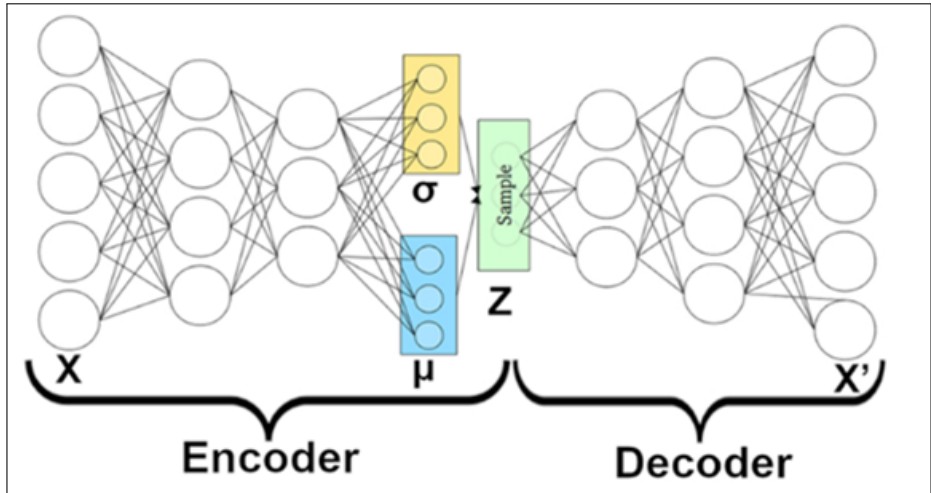

Figure 2: Variational Auto-Encode model.

The VAE model can be presented by the expressions:

$$\begin{cases} \mu = f(\mathcal{X}) \\ \sigma = (\mathcal{X}) \\ \mathcal{X}' = g(\mathcal{Z}) \\ \mathcal{Z} = \mu \oplus \sigma \otimes \epsilon \\ \epsilon \sim \mathcal{N}(0, \mathcal{I}) \\ \mathcal{Z} \sim \mathcal{N}(\mu, \sigma^2) \end{cases} \tag{1}$$

where $\mu$ is the vector of means, $\sigma$ denotes the vector of the standard deviations, and $\epsilon$ designates a small constant equal to the value of the reduced normal distribution $\mathcal{N}(0, \mathcal{I})$, $Z$ corresponds to the latent space, $X$ refers to the input image and $X'$ is the reconstructed image. The encoder is $f$, while the decoder is $g$. The VAE model aims to minimize the loss function $L$ by optimizing the two functions: , $f$ (encoder) and $g$ (decoder). $L$ comprises the regularization loss ($KL$), which is the kullback-Leibler divergence loss, and reconstruction loss ($RL$).

$$L = RL + KL \tag{2}$$

The explanation for the reconstruction loss $RL$ is:

$$RL = 1/2|X - X'|^2 \tag{3}$$

The explanation for the kullback-Leibler divergence loss $KL$ is:

$$KL = DKL(q(X)||p(Z))) = DKL(N(\mu, \sigma^2)||N(0, I)) = \frac{1}{2}\sum_1^n (\log(-\sigma^2) + \log \mu^2 + \log \sigma^2 + 1) \tag{4}$$

And by incorporating (Eq3) and (Eq4) into (Eq2), our model's loss function $L$ becomes:

$$L = 1/2|X - X'|^2 + \frac{1}{2}\sum_1^n (\log(-\sigma^2) + \log \mu^2 + \log \sigma^2 + 1) \tag{5}$$

The term $n$ represents the hidden space dimension $n=\dim(Z)$.

### 3.2 SIAMESE VAE BACKGROUND

A Siamese network is an architecture containing two parallel neural networks having identical configurations with identical parameters and shared weights. The suggested architecture is made up of two VAEs connected by a vector of features created by concatenating the two latent spaces of the

pair VAE and a classification component trained with a source ground truth map ($C_s$). The loss function $LS$ of the developed Siamese VAE is composed of the loss function of $VAE_1$ ($L1$), the loss function of $VAE_2$ ($L2$) and the cross-entropy loss ($CE$) of the classification part.

$$LS = L1 + L2 + CE \tag{6}$$

The explanation for the cross-entropy loss $CE$ is:

$$CE = -\sum_{i=1}^{c} T_i \log Y_i \tag{7}$$

Where Y is the result, $c$ is the number of classes, and $C$ is the ground truth input to the classification component. The loss of our model becomes.

$$L_{VAEDesert} = 1/2|X_1 - X'_1|^2 + 1/2|X_2 - X'_2|^2 + \sum_{1}^{n}(\log{(-\sigma_1^2)} + \log{\mu_1^2} + \log{\sigma_1^2} + 1)$$

$$-\sum_{i=1}^{c} T_i \log Y_i \tag{8}$$

### 3.3 TRANSFER LEARNING

We consider a source domain $D_s$ consisting of the source scene $X_s$, containing the ground truth $C_s$, and the target domain $D_t$ made up of the target scene $X_t$ containing the ground truth $C_t$ and the unknown knowledge to be learned. The knowledge learned in the source domain can be transferred to the target domain through TL. TL applies the knowledge in $D_s$ to help learn the knowledge in $D_t$. In this study, the condition $D_s \neq D_t$ and $C_t$ are always smaller, compared to $C_s$. In the learning process, the classification knowledge of the source scene $X_s$ is employed to detect the classification of the target scene $X_t$. Besides, TL is used to address the classification issue by utilizing the fine-tuning technique that can be divided into freezing and retraining, as shown in Figure 3.3. Firstly, feature extraction and classification are applied using a pre-trained Siamese VAE on the source scene. Then, the target Siamese VAE model is initialized with the saved weights of the source model which is the freezing part. Finally, the ground truth of the target scene is used to retrain the target model and predict the classification map.

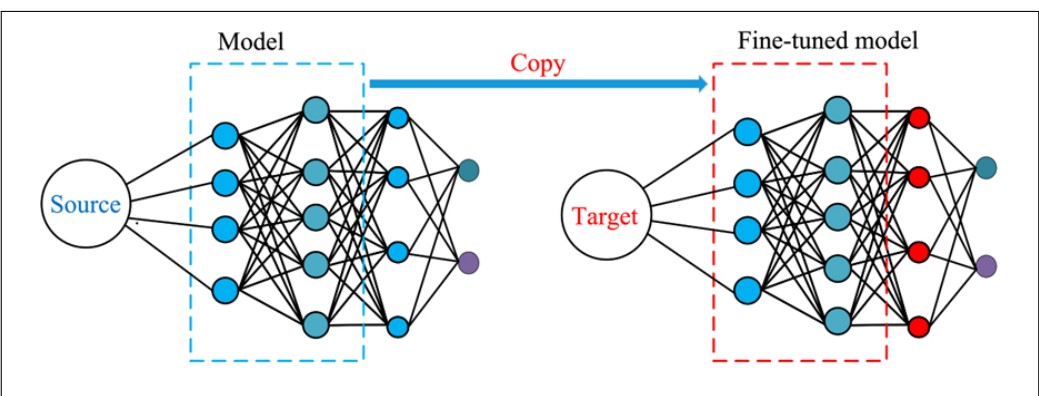

Figure 3: TL with model fine-tuning adapted (Wang et al., 2021).

## 4 EXPERIMENTAL EVALUATION

Recently, computer vision techniques have been widely employed in environmental monitoring due to the significant improvements in machine learning algorithms and the increasing availability of free remote sensing data with moderate spatial and temporal resolution. Enhancing the DL models ability to monitor the spatial and temporal distribution of desertification is highly recommended to

ameliorate combating desertification operations and ensure ecosystem security. In the present work, the introduced Siamese VAE method is applied with TL strategy to detect desertification risk in two study areas.

### 4.1 STUDY AREA AND DATASET

Each of the considered scenes of Landsat images is composed of three bands of the Landsat visible spectrum (RGB) and the normalized vegetation index $NDVI$. To calculate the Normalized Difference Vegetation Index $NDVI$, the reflectance of the red ($R$) and near-infrared ($PIR$) channels, measured in the visible band, were used. The $NDVI$ calculation formula is written below.

$$NDVI = \frac{PIR - R}{PIR + R} \tag{9}$$

Bi-temporal Landsat images were acquired respectively from Menzel Habib, Gabes located in the south of Tunisia, as the source scene, and from Jeffara, Medenine, in the south of Tunisia, as the target scene from the ClimateEngine website. Figure 4.1 illustrates the two study areas. Each pair of images was acquired from the Landsat project in 2001 and 2004 with a spatial resolution of 30 meters. Desertification risk information of both image pairs was utilized to verify the accuracy of the produced maps.

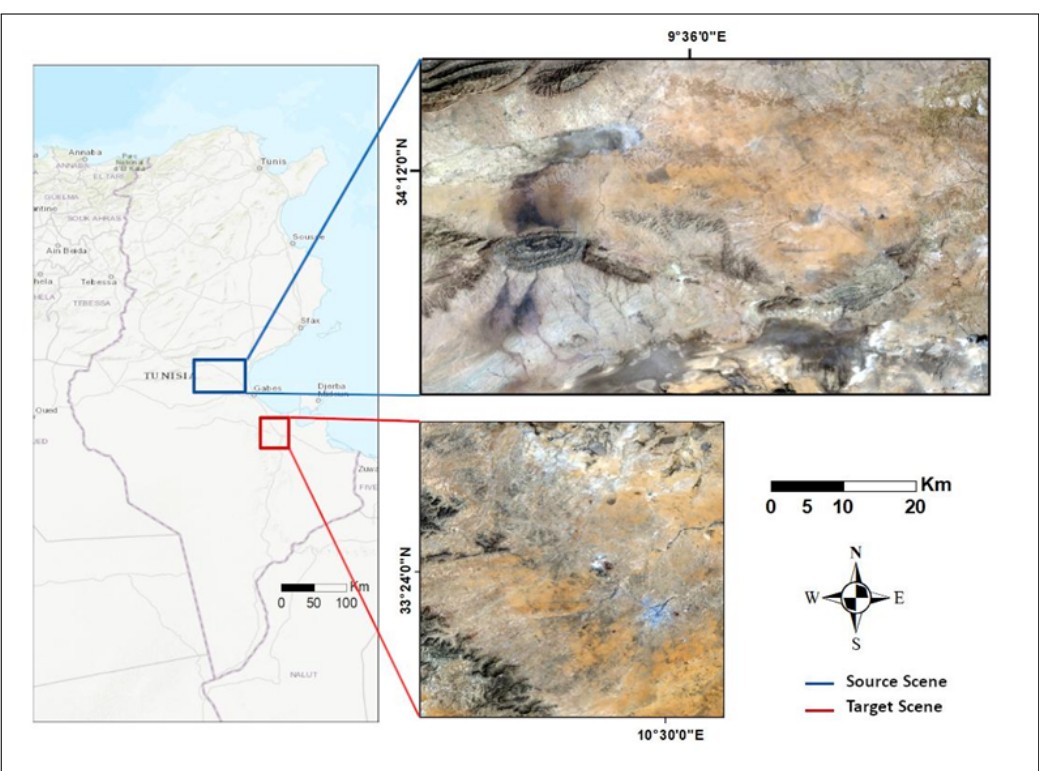

Figure 4: Study Areas: Source and Target Scenes.

### 4.2 IMPLEMENTATION DETAILS

The proposed method was implemented using Keras (Ketkar, 2017). The categorical cross-entropy loss was used as the loss function for the classification (Kingma & Ba, 2014), while stochastic gradient descent (Adam) (Kingma & Ba, 2014) was employed as the optimizer. The batch size was set to 256 and the number of the training data in the source scene was more than 500 000 and about 90 000 in the target scene. The introduced model was applied to train 100 epochs. The learning rate was set to 0.001 and the latent space dimension was equal to 24, as demonstrated in Table 1.

Table 1: Specifics of the grid search considered by each method. The ideal setting for each hyperparameter is indicated with a bold font.

| Method | Searching Range of hyperparameters |
|---|---|
| Siamese VAE source Scene | learning rate: [0.0001, **0.001**, 0.01], latent space dim [2,4,8,16,**24**,32], batch size [64,128,**256**,512,1024], epoch [10,50,**100**,200] |
| Siamese CNN source Scene | learning rate: [0.0001, **0.001**, 0.01], hidden space dim [2,4,8,16,**24**,32], batch size [64,128,**256**,512,1024], epoch [10,50,**100**,200] |

### 4.3 EVALUATION METRICS

Desertification detection results were compared with the reference data per pixel to assess quantitatively the efficacy of the developed method in terms of $Accuracy$, recall ($R$), precision ($P$) and $F1 - score$. The classification Accuracy was obtained by dividing the total number of correct predictions by the total number of predictions. However, recall measured the number of positive class predictions made out of all positive examples, as demonstrated in Eq.10.

$$R = \frac{TP}{TP + FN} \tag{10}$$

where true-positive ($TP$) and true-negative ($TN$) denote the correct number of changed and unchanged correctly detected pixels. In fact, false-positive ($FP$) and false-negative ($FN$) denote the number of the changed and unchanged incorrectly detected pixels, respectively. On the other hand, precision quantifies the number of positive class predictions, as revealed in Eq.11.

$$P = \frac{TP}{TP + FP} \tag{11}$$

The $F1 - score$ is a comprehensive evaluation index expressed by the average harmonic value of precision and recall, as illustrated in Eq.12.

$$F1 - score = \frac{2 * Precision * Recall}{Precision + Recall} \tag{12}$$

Generally speaking, a method is considered more efficient if it provides higher $Accuracy$, $P$, $R$, and $F1 - score$. Since the $Accuracy$ and the $F1 - score$ are the most important metrics in the multi-classification task, they were used in the experiments to select better deep learning-based methods.

## 5 RESULTS AND DISCUSSION

### 5.1 COMPARATIVE METHOD EVALUATION

The Siamese VAE was compared to SCNN, with and without the application of TL strategy, to demonstrate its better efficiency. In the conducted experiments, the same number of training samples was used. More than 500000 labeled pixels from the source scene and around 90000 labeled pixels were extracted from the target scene to make up the ground truth data. Table 2 displays the outcomes of the Siamese VAE and SCNN applications on the ground truth data. The obtained results show that the two methods (Siamese VAE and SCNN) perform well in desertification detection, the standard deviations were stable and only varied within a small range. More precisely, the Siamese VAE provided the best Accuracy , R, P and F1-scores in source and target scenes. It also ensured better feature extraction thanks to: 1) the normal distribution (Gaussian) of the latent space by the VAE, and 2) the spatial consistency conservation of the extracted features obtained by the Siamese structure.

### 5.2 DESERTIFICATION DETECTION OUTCOMES

The distribution of desertification (Figure 5.2) along the two study areas during the period from 2001 to 2004 demonstrates that the desertification areas (high and very high risk) were primarily located

Table 2: Quantitative results on the Gabes dataset.

| Model | Accuracy | Precision | Recall | F1-score |
|---|---|---|---|---|
| Siamese VAE Source | **98,16** | **98,15** | **98,12** | **98,14** |
| Siamese VAE Target with TL | **92,41** | **91,29** | **91,06** | **90,84** |
| Siamese VAE Target without TL | 91,16 | 89,64 | 89,60 | 89,62 |
| SCNN Source | 96,98 | 96,85 | 96,55 | 96,70 |
| SCNN target with TL | 90,07 | 88,41 | 88,82 | 88,61 |
| SCNN target without TL | 89,11 | 88,14 | 89,83 | 88,90 |

in the lowland regions characterized by intensive agricultural activities. In fact, the latter together with unfavorable meteorological conditions have intensified the desertification risk in the two study areas. The most important agricultural activity in the two studied regions was rainfed and irrigated olive. The source scene's desertification could be more effectively detected using the Siamese VEA approach. The desertification detection was enhanced and refined in the target scene by applying the TL strategy.

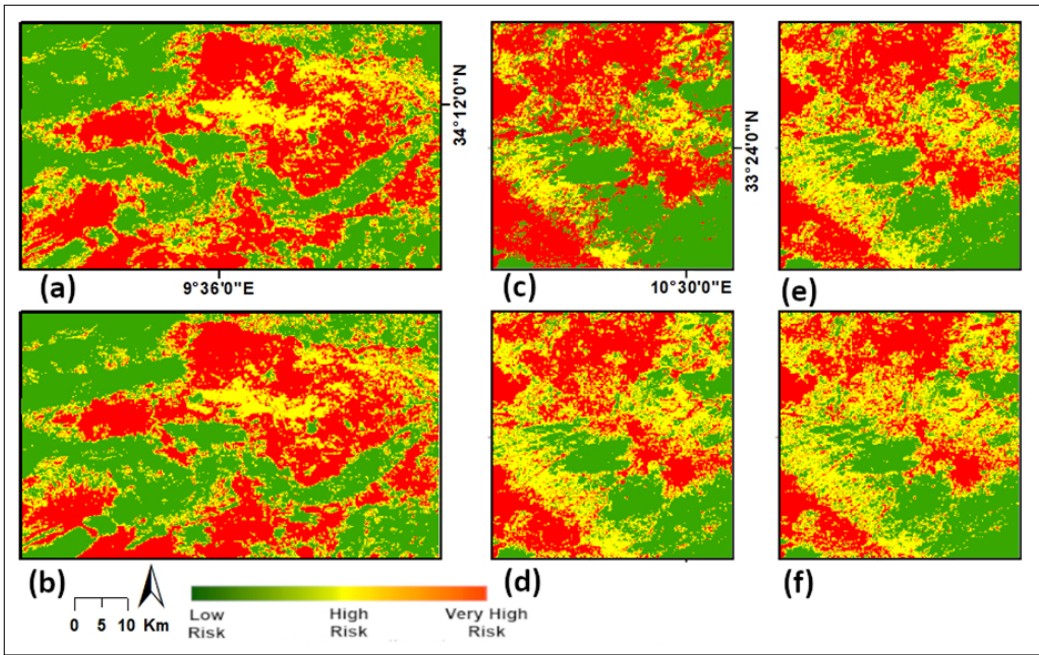

Figure 5: Desertification detection outcomes: (a): Siamese VAE prediction in source Scene, (b): SCNN prediction in Source Scene, (c): Siamese VAE prediction in Target Scene with TL,(d): SCNN prediction in Target Scene with TL (e): Siamese VAE prediction in Target Scene without TL, (f): SCNN prediction in Target Scene without TL.

## 5.3 THE ADVANTAGES AND LIMITATIONS

In two dry areas of southern Tunisia, computer vision based on the Siamese DL model and TL strategy were used to create a desertification risk map with a high spatial and temporal resolution from bi-temporal Landsat images. The obtained findings demonstrate that the developed method was efficiently employed to detect the land surface change and monitor desertification. Obviously, TL was more sophisticated than a single method Siamese VAE or SCNN and it showed higher performance in the target scene with less ground truth data. It is also clear that, in the target scene, the accuracy of both Siamese methods was less than 93%, depending on the quantity and quality of the labeled data. These results may impede the identification of the desertified areas and alter

the outcomes of desertification monitoring. To produce more reliable results, In future studies, the labeled data in the target scene should be improved in quantity and quality.

## 6 CONCLUSION AND OUTLOOK

This work introduced a classification method based on Siamese VAE and TL for CD in bi-temporal landsat images. The Siamese network trained in a labeled source scene was transferred to the bi-temporal image of the target scenes with less labeled data using a fine-tuning technique. Two datasets, including Landsat bi-temporal images collected in Tunisia's southern, arid regions of Gabes and Medenine, were processed. The experimental results revealed that the proposed method is more efficient than SCNN in terms of CD and classification. As future research perspectives, the appropriate classification in the target scene can be investigated by applying an active learning approach for the optimal selection of labeled data. Additionally, we intend to test the performance of the suggested method when using more input data and applying an active learning strategy to improve the quality of the ground truth data in the target scene.

## ACKNOWLEDGMENTS

We want to thank the Institute of Arid Regions of Medenine, Medenine, Tunisia, laboratory LESOR (laboratory of economics and rural societies), for providing the ground truth data.

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
