# OpenReview forum: "Change Detection for bi-temporal images classification based on Siamese Variational AutoEncoder and Transfer Learning"
_ICLR.cc/2023/Conference — Submitted to ICLR 2023_

### Official Review · Reviewer_Zx8V · 2022-10-20

**Confidence:** 5
**Clarity, Quality, Novelty And Reproducibility:** 1. The writing and organization of th…
**Correctness:** 3
**Technical Novelty And Significance:** 1
**Empirical Novelty And Significance:** 1
**Recommendation:** 3

**Strength And Weaknesses:**

Strength: The proposed method is reasonable and feasible.
Weaknesses: However, the novelty of this work is limited. Siamese networks and VAE are common techniques for many applications. The authors only combine them simply to deal with their tasks. I cannot any innovative ideas in the current manuscript. Also, the testing data are remote sensing (RS) images. Nevertheless, the proposed method looks like a general-purpose model. Do the author sure that their method is suitable for RS images?


**Summary Of The Paper:**

This paper presents a new method for bi-temporal images classification based on Siamese variational autoencoder (VAE) and transfer learning (TL). First, the suggested generative method utilizes two VAEs to extract features from bi-temporal images. Then, the obtained features are concatenated into a feature vector. To get a classification map of the source scene, the classifier receives the concatenated vector and the ground truth data as input. The source model is fine-tuned to be applied to the target scene with less ground truth data using a TL strategy.

**Summary Of The Review:**

Based on the comments displayed above, I do not recommend this work to be published in ICLR.

---

### Official Review · Reviewer_CkES · 2022-10-24

**Confidence:** 1
**Clarity, Quality, Novelty And Reproducibility:** Please I will provide the review on W…
**Correctness:** 1
**Technical Novelty And Significance:** 1
**Empirical Novelty And Significance:** Not applicable
**Recommendation:** 1

**Details Of Ethics Concerns:**

Please I will provide the review on Wednesday.

**Strength And Weaknesses:**

Please I will provide the review on Wednesday.

**Summary Of The Paper:**

Please I will provide the review on Wednesday.

**Summary Of The Review:**

Please I will provide the review on Wednesday.

---

### Official Review · Reviewer_8FMz · 2022-10-24

**Confidence:** 4
**Correctness:** 3
**Technical Novelty And Significance:** 2
**Empirical Novelty And Significance:** 2
**Recommendation:** 3

**Clarity, Quality, Novelty And Reproducibility:**

The article is well written and the proposed approach is easy to follow. The novelty is limited since other very similar approaches have been proposed on the literature before and the limited evaluation presented make it difficult to understand the performance of the proposed approach compared to other apporaches. The work can be reproduced to some extent using provided details.

Some questions:
1. Was the classifiers trained jointly with the siamese autoencoder as well?
2. How well does the classifier performs if trained on the input images instead of the latent space from the autoencoder?

**Strength And Weaknesses:**

-- Strength:
- Proposed idea is technicaly sound.
- This work has the potential of being impactfull since accurate change detection approaches are important for multiple earth Observation applications

-- Weaknesses:
The work is missing exhaustive and proper evaluation.
- The proposed approach does not include any wellknown benchmark change detection dataset beyond their own sentinel 2 scenes from South Tunisia. I list below a few baseline datasets to consider. Many more are available in the article by Shi et al in 7.
- Their results are only compared to a Siamese CNN architecture ignoring multiple other proposed architecture like the one proposed in 1.

Change detection dataset:
- CDD dataset from 4.
- Onera Satellite Change Detection dataset from 5
- xBD dataset from 6

*Related Work to consider:*
1. Chen, J., Yuan, Z., Peng, J., Chen, L., Huang, H., Zhu, J., ... & Li, H. (2020). DASNet: Dual attentive fully convolutional Siamese networks for change detection in high-resolution satellite images. IEEE Journal of Selected Topics in Applied Earth Observations and Remote Sensing, 14, 1194-1206.
2. Daudt, R. C., Le Saux, B., & Boulch, A. (2018, October). Fully convolutional siamese networks for change detection. In 2018 25th IEEE International Conference on Image Processing (ICIP) (pp. 4063-4067). IEEE.
3. Mesquita, D. B., dos Santos, R. F., Macharet, D. G., Campos, M. F., & Nascimento, E. R. (2019). Fully convolutional siamese autoencoder for change detection in UAV aerial images. IEEE Geoscience and Remote Sensing Letters, 17(8), 1455-1459.
4. Lebedev, M. A., Vizilter, Y. V., Vygolov, O. V., Knyaz, V. A., & Rubis, A. Y. (2018). CHANGE DETECTION IN REMOTE SENSING IMAGES USING CONDITIONAL ADVERSARIAL NETWORKS. International Archives of the Photogrammetry, Remote Sensing & Spatial Information Sciences, 42(2).
5. Rodrigo Caye Daudt, Bertrand Le Saux, Alexandre Boulch, and Yann Gousseau, “Urban change detection for multispectral earth observation using convolutional neural networks,” in International Geoscience and Remote Sensing Symposium (IGARSS). IEEE, 2018
6. Gupta, R.; Goodman, B.; Patel, N.; Hosfelt, R.; Sajeev, S.; Heim, E.; Doshi, J.; Lucas, K.; Choset, H.; Gaston, M. Creating xBD: A dataset for assessing building damage from satellite imagery. In Proceedings of the IEEE Conference on Computer Vision and Pattern Recognition Workshops, Long Beach, CA, USA, 16–20 June 2019; pp. 10–17.
7. Shi, W., Zhang, M., Zhang, R., Chen, S., & Zhan, Z. (2020). Change detection based on artificial intelligence: State-of-the-art and challenges. Remote Sensing, 12(10), 1688.


**Summary Of The Paper:**

Authors proposed the use of a siamese variational autoencoder (SVAE) for change detection over bi-temporal imagery. The model receives as input to scenes of the same spatial location but collected during different times. The SVAE is optimized using a crossentropy classification loss computed using an image classifier on top of the concatanation of the latent representations of the two input images in addition to  standard reconstruction losses. The model was evaluated for the task of desertification detection using Sentinel 2 imagery in South Tunisia. The model shows better performance than siamese convolutional neural networks for this task.

**Summary Of The Review:**

The paper is well written and easy to follow, but need further evaluation to better attest the performance of the proposed framework. Comparison to other approaches using change detection benchmark datasets would improve paper quality.

---

### Official Review · Reviewer_G4jm · 2022-10-25

**Confidence:** 3
**Correctness:** 1
**Technical Novelty And Significance:** 1
**Empirical Novelty And Significance:** Not applicable
**Recommendation:** 3

**Clarity, Quality, Novelty And Reproducibility:**

- As stated in the weaknesses, I am concerned about the limited novelty of the proposed method. Besides, the paper is presented tediously, the readers, especially who are unfamiliar with the topic, may find it hard to follow.
- Reproducibility may also be a concern, as the authors do not provide source code and dataset.


**Strength And Weaknesses:**

Strengths:

The paper proposes to adapt Siamese VAE and Transfer Learning on top of a CNN network, which shows improvement over the baseline on a dataset of Landsat visible spectrum and normalized vegetation index collected in Tunisia.


Weaknesses:

Overall, the paper presentation is tedious. The Introduction and the Method sections do not clearly present what problem the paper is trying to solve, which creates challenges for readers to understand the context of the paper.
In Related Works, the authors did not show the differences between their method and the previous works.
I am concerned about the limited novelty of the proposed method. The author seems trying to collect several popular methods from related fields into the paper (e.g., Siamese VAE, transfer learning) without explicitly explaining the intuitions, the reasons, and the problem that leads authors to use those methods to fix them.
Although there are improvements reported in Table 2, the gap seems to be small (<2%) on most of the metrics when comparing respective rows (prefixes of Siamese VAE vs. SCNN). Notably, the performances are very high even with target scenes (almost 90%).

**Summary Of The Paper:**

The paper tackle problem of change detection from bi-temporal images by proposing to adapt a Siamese VAE method and Transfer Learning.

The performance of the proposed method on the bi-temporal image dataset collected in South Tunisia improves over a baseline of Siamese Convolutional Neural Network

**Summary Of The Review:**

As there are quite many problems in the presentation of the paper, as well as limited, unclear novelty and reproducibility, I would not recommend this paper to be accepted.

---

### Decision · Program_Chairs · 2023-01-20

**Decision:**

Reject

**Justification For Why Not Higher Score:**

The paper is subpar technically, and it is impossible to tell if the proposed method adds value compared to previous similar work. The main cons are the lack of novelty and insufficient experimental evaluation, in terms of the dataset used and the related works compared to.

**Justification For Why Not Lower Score:**

N/A

**Metareview: Summary, Strengths And Weaknesses:**

The paper proposes a method for change detection based on VAEs and evaluates it on a custom dataset.

All reviewers agree that the paper is not fit for publication at this point (the review by Reviewer CkES is discarded since it is very brief and delivered in a non-standard way). The main cons are the lack of novelty and insufficient experimental evaluation, in terms of the dataset used and the related works compared to.

I recommend rejection at this point, since the paper, while addressing an important problem, is subpar technically, and it is impossible to tell if the proposed method adds value compared to previous similar work.